# Valuable Cheap Talk and Equilibrium Selection

**Julian Jamison**

Department of Economics, University of Exeter Business School, Rennes Drive, Exeter EX4 4PU, UK;
J.Jamison@exeter.ac.uk

**Abstract:** Intuitively, we expect that players who are allowed to engage in costless communication before playing a game would be foolish to agree on an inefficient outcome amongst the set of equilibria. At the same time, however, such preplay communication has been suggested as a rationale for expecting Nash equilibria in general. This paper presents a plausible formal model of cheap talk that distinguishes and resolves these possibilities. Players are assumed to have an unlimited opportunity to send messages before playing an arbitrary game. Using an extension of fictitious play beliefs, minimal assumptions are made concerning which messages about future actions are credible and hence contribute to final beliefs. In this environment, it is shown that meaningful communication among players leads to a Nash equilibrium (NE) of the action game. Within the set of NE, efficiency then turns out to be a consequence of imposing optimality on the cheap talk portion of the extended game. This finding contrasts with previous "babbling" results.

**Keywords:** strategic communication; two-stage games; pareto efficient equilibria; belief formation

## 1. Introduction

Self-enforcing agreements—those for that no party has any incentive to break given that all others comply—should be carried out even if they are not binding in a formal sense. This is in fact the defining characteristic of the standard Nash equilibrium concept, and thus, one of the common justifications for this concept is that if players are allowed to communicate before playing a game, they could hardly reasonably agree on an outcome not satisfying this criterion. Recall that a Nash equilibrium constitutes for each player a set of strategies and beliefs (about other players' strategies), such that the strategies are the best responses to beliefs and the beliefs are correct (see e.g., Osborne 2004 [1]). We assume that there is no recourse to court-enforceable contracting, or equivalently that any such interactions have already taken place. Unfortunately, while intuitively pleasing, this justification for the use of a Nash equilibrium has been characterized by a shortage of formal models.

On a related, but distinctly different track of reasoning, it is natural to wonder why agents would ever agree on an inefficient outcome, assuming that they had the chance to talk in the first place. In other words, why would players agree ahead of time to an inefficient outcome of a game if there were another potential outcome, also an equilibrium, that gave strictly greater payoffs to all of them? Once again, the challenge has lied in constructing a realistic, but necessarily simplified, formal model of the agents' communication process. Among other problems, this inefficient result appears to be incompatible with the arguments outlined above, in which Nash equilibria in general are justified.

This type of preplay communication is often called *cheap talk*, which may be roughly defined as nonbinding, nonpayoff relevant, preplay communication. Although cheap talk has indeed received attention as a potential solution to these questions surrounding the equilibrium concept, in practice, it has been mostly used in the study of signaling games, in repeated environments (often in connection with learning), and in certain applied settings. These are of course all important applications, but these leave the original ambiguities unresolved. This paper, then, returns to the goal of constructing a

more comprehensive model of pure cheap talk and explores its relationship with equilibria and equilibrium selection.

This paper develops a model of cheap talk that involves an unlimited communication session, called a *conversation* , before the play of a standard game begins. Players announce in advance what actions they plan to take in the upcoming game, and taken together, these announcements form one possible prediction of what they may actually do. On the other hand, there is also a common prior forecast, given exogenously, of what each player will do; this forecast is updated as the conversation proceeds. An announcement is defined to be credible only if it is close to the best response to one or the other of these two predictions about the rest of the players. Otherwise, an announcement has no external justification, so it is deemed unbelievable and disregarded. The conversation continues indefinitely in this manner, possibly, but not inexorably toward some limit. Realistically, it will rarely if ever go on for very long (although, for complicated games, it may take some time), since if it is going to converge, it will do so rapidly. However, it is important not to have an artificial limit imposed externally, just as long finitely repeated games behave very differently than infinitely repeated games.

The paper's first main result is that if the conversation converges toward a limit, then this limit must be a Nash equilibrium of the underlying action game in which payoffs are determined.[1] Conversely, any Nash Equilibrium forms a possible limit of the conversation. This result can be interpreted as saying that meaningful communication before a game can only lead to Nash outcomes. Since the cheap talk is the initial interaction between the players, we assume that they cannot be sure of the strategies that their opponents will follow in the communication stage. Any strategy in this phase that is weakly dominated by another is clearly not optimal; anything else is potentially the preferred choice and is therefore, given the lack of information, one possible optimal choice.[2] The paper's second result then states that optimal pregame play in the conversation stage leads to an efficient outcome and that any efficient final outcome is a possible result of such a strategic conversation.[3] This can be interpreted as saying that rational, or thoughtful, speech leads to efficiency. This completes the connection between cheap talk (as modeled here, i.e., in an environment where rationality and utilities are common knowledge), Nash equilibria, and Pareto optimality. The first result applies to all games (at least those with a Nash equilibrium), while the second result only has bite in games with multiple equilibria.

The conclusion derived from the second main result contrasts with previous "babbling" results, in which it is impossible to select among the set of Nash equilibria because all pregame communication is ignored. The main reason for the difference is that those previous studies looked for equilibria of the extended communication game as a whole—for instance, by assuming that the full strategies of all players are known. This allows equilibrium strategies in which no value is placed even on seemingly mutually informative communication, whereas the model below presupposes the impossibility of ignoring beneficial interchange. Thus, the present paper takes a more primitive view of pregame strategies, especially since in part it is attempting to justify the equilibrium concept in the first place. Naturally, although the model does not impose beliefs about the cheap talk stage, it still must make some assumption about beliefs held upon entering the action game. Another approach that will destroy the babbling equilibria is to assume an arbitrarily small, but positive cost to sending messages—this is a restriction on the environment rather than on the structure of equilibrium or on belief formation. While this limitation is plausible in reality, it is, strictly speaking, no longer a model of cheap talk, even if the total sum spent on sending messages is always lower than the game's smallest payoff differential.

The paper proceeds as follows. Section 2 provides a brief survey of some of the relevant literature. In Section 3, some motivation is given for the specific assumptions made in this conversational model

---

1  The limit is an $\varepsilon$-Nash equilibrium.
2  This is discussed in further detail in Section 3.
3  The notion of efficiency used here is *stable efficiency*, a concept that is equivalent to Pareto efficiency in generic two person games.

of cheap talk. Section 4 lays out the formal model, stating and proving the paper's two main results. Several examples are detailed in Section 5 in order to illustrate both the cheap talk process and the implications of the theorems. Finally, Section 6 concludes the paper by summarizing the model and discussing some possible extensions of its implications.

## 2. Previous Literature

The concept of cheap talk was introduced into the economics literature by Crawford and Sobel (1982) and Farrell (1987) [2,3]. Since then, a sizable literature has developed related to this topic, with such examples as Farrell and Gibbons (1989), Forges (1990), Farrell (1993), Aumann and Hart (1993), Blume and Sobel (1995), and a survey in Farrell and Rabin (1996) [4–9]. The paper that perhaps is closest to the present one is Rabin (1994) [10]. It models a finite instead of an infinite opportunity for communication, but also seeks a notion of optimality rather than equilibrium in the analysis of the extended game. The specific form of cheap talk assumed by Rabin is different from the one presented below, in particular with respect to the element of choice between strategies against which to credibly best respond. The results can be framed in terms of the two central questions posed here, but are generally less conclusive in either. Both papers adhere to the full rationality paradigm of classical game theory and previous work on cheap talk, as opposed to, say, the evolution literature.

There are a number of papers that study a more limited class of games. For instance, Matsui (1991) [11] applied cheap talk to common interest games, and in this context, his notion of *cyclic stability* yields efficiency. Canning (1997) [12] studied signaling games of common interest, although the messages do not necessarily constitute cheap talk per se. He found that off-path beliefs are vital to the question of whether or not efficiency is eventually realized; randomly drawn off-path beliefs encourage experimentation and lead to efficiency. Finally, Sandroni (2000) [13] studied two person repeated coordination games without cheap talk. He introduced the concept of *blurry beliefs*, which is a less restrictive (that is, more fully rational) belief dynamic than those used in evolutionary game theory, although it is stronger than anything used here. Sandroni showed that if the belief classes of the players satisfy *reciprocity*, then cooperation will be achieved. Overall, the current paper pins down the link between communication and (efficient) equilibrium outcomes more concretely than the previous literature. In particular, it explores a specific empirically-consistent model of belief formation and shows a two way equivalence between that process and the optimality of the resulting behaviors.

A fairly large class of papers has studied repeated games and the emergence of Nash equilibria without introducing cheap talk, including Crawford and Haller (1990), Young (1993), and Kalai and Lehrer (1993) [14–16]. Finally, there have been some experimental studies of communication and equilibrium selection in various coordination games; see, for example, Cooper et al (1992), Brandts and Cooper 125 (2007), and Cachon and Camerer (1996) [17–19]. The results can be summarized (and oversimplified) as finding that two way pregame communication greatly increases the chances of observing efficient equilibrium outcomes. Pertinently, this holds even if the efficient equilibrium is not risk-dominant, in contradistinction to some previous results. Meanwhile, some experimental studies found that preplay communication can actually induce fewer choices consistent with Nash equilibria, e.g., Boulu-Reshef et al (2020) [20] in the context of public goods games. This could either be due to the limited opportunity for communication and/or the possibility of social preferences (which would change the set of NE).

## 3. Motivation

This section provides some intuition and justification for the structure of the model that follows; the impatient reader can skip to the next section. The model assumes that there is an action game to be played, about which the players are assumed to have full information (in order to abstract away from any signaling incentives during the conversation). Each player begins with a common forecast about what actions he or she will take in the upcoming game. These expectations can be interpreted as vague initial ideas about how the game might be played, arising perhaps from societal conventions or from

focal points (hence the assumption that the forecasts are common and known). They are not beliefs in the formal sense, although they will be updated throughout the conversation.[4] Since a priori, nothing can be absolutely ruled out by any of the players, the prior forecasts are totally mixed.[5] Needless to say, the forecasts are not in any way binding: players ignore what they themselves are "expected" to do, although they can take into account the influence this expectation has on their opponents.[6] The key distinction between forecasts and standard beliefs is that the forecasts are about the general environment (how might this game typically be played by others?), whereas beliefs are about the actual decisions by the specific players interacting in a given concrete situation. Thus, among other implications, it makes sense to reason about players trying to influence the beliefs that their opponents have about them, whereas they cannot influence the more broadly prevalent forecasts. Of course, then, we need to model from whence the forecasts come (social norms, news media, evolutionary psychology, etc), but that is outside the scope of the present paper.

During the conversation stage, before playing the action game, players send public messages to each other. Since we are attempting to understand what such preplay communication can achieve, we assume that there is an unlimited (but countable) opportunity to send these messages. For simplicity and without loss of generality, the messages are taken simply to be announcements of a player's own expected actions in the game. One could assume instead that players announce mixtures of their possible actions, but this is an unnecessary complication. Essentially, given infinite riskless communication, this slight limitation on the flexibility of messages imposes no loss in the long run. Implicitly, we are assuming that players can understand one another and that they take messages at face value (not in a strategic sense, but in a linguistic sense). If the message "action L" is sent, everyone understands that to mean "action L" and not "action R". Thus, there is a *natural language* for speech; the players share enough common history or cultural affiliation that they are able to talk and understand one another in a previously unencountered situation.

Of course, not all announcements should be considered seriously. We need to define a notion of *credibility* or believability. The first requirement is that a player's announced action should be *self-committing*, in the sense that if it were believed and best responded to, the original announcer would still be willing to carry through with it (within the confines of the action game). This requirement is equivalent, then, to being in the support of some Nash equilibrium of the action game. At the beginning of the preplay conversation, any self-committing action is credible, so players have a chance to guide the discussion. In general, there will be some tradeoff between allowing the players leeway to influence the conversation at the beginning, but requiring them at some point to pay attention to what the others are saying and to reflect that updated information in their own announcements. Unlike in the deterministic best response dynamics of evolutionary models, it is important in this model that players have a choice over what to say; this is the hallmark of a conversation. It is this choice, along with the lack of payoffs until the action game is at the very end, that differentiates this paper's model from an evolutionary learning model.

The common forecast is very slowly updated by each credible announcement. We can think of the prior forecast as the result of a long, but finite fictitious history of credible announcements, with each new stated action adding to the average.[7] As beliefs get updated, the initial forecast can be ignored and only the actual credible announcements counted toward an average forecast: this forecast constitutes a

---

[4]　The players do not have beliefs about the full strategies of their opponents, only ideas about what might actually occur in the game. Thus, the preplay forecasts are distributions over actions, not distributions over mixed strategies (which themselves are distributions over actions). This is not crucial to the conclusions reached.

[5]　It is not strictly necessary for the results that the priors be totally mixed.

[6]　The author performed the analysis under the seemingly weaker assumption that all that is known about the prior forecasts is that they place a certain minimum weight on each action, but the results carry over. Since this assumption adds complexity, but is no sounder in justification (Why can the entire distributions not be known if the minimum weights are?), it has been left out.

[7]　Recall that the average of multiple sets of actions is equivalent to a mixed strategy.

player's *appearance*. In general, we recursively define an announced action to be credible if it is the best response (within $\varepsilon$) to either the current forecast of an opponent's behavior or to an opponent's appearance.[8] If there are more than two players, either the common updated forecast or a player's appearance may be substituted for each. The intuition here is that a player can either say something like, "This is what I think you are going to do, and if so then I would plan to do such-and-such," or something along the lines of, "Okay—for the moment I'll take you at your face value, and in that case I'll want to do so-and-so." Of course, he or she only needs to consider credible announcements in making these plans.

At any time during the preplay conversation, a player can make any announcement desired, but only those statements that are credible will have an impact on the conversation. Since all players know the prior forecasts and all previous announcements, they can calculate which of these announcements were actually credible and hence also which of their own announcements will be perceived as credible by others. If at any point, there is but a single action that is credible for a particular player, it must be that this player can only seriously be considering that action (at that point in time). Therefore, in effect, it does not matter whether or not he or she actually announces that action; everyone knows that it is being considered, and hence, it should count toward the forecast and appearance of that player, regardless of what may or may not be announced. This argument implies that without loss of generality, we may assume that all players make credible announcements during each round of the conversation.[9] Finally, we assume that at each point in time, any player can start over; that is, declare a clean slate and remake their appearance anew. This is the equivalent of declaring that the conversation has broken down from his or her perspective and, among other implications, allows the players to attempt to coordinate. Although it may seem like an overly strong possibility, in fact, a player's appearance is a powerful commitment device, and so, giving up on it involves a significant loss.[10] In any case, of course, the clean slate option is available equally to all of the players. This completes the description of the cheap talk conversation.

One last remaining question about the credibility concept concerns the infinite durability of credible announcements. That is, a credible announcement always "counts" even if it is no longer credible. The reason for this is that any credible announcement indicates evidence of a desire for that action if possible, and there is no reason to think that the desire will change or that the desired action may not once again become plausible. In effect, each announcement has a small impact that builds toward the whole impression, rather than the fads of currently credible actions. In fact, if only those actions that are credible at the moment are averaged into the player's appearance, at each communication stage, one can observe swings back and forth of what is and is not believable. Furthermore, in this updated setting, eventually, only one pure strategy will be credible, and so, it is essentially impossible for players to converge to a mixed strategy.

Once the preplay conversation is complete, we have a countably infinite sequence of announcements for each player, with an associated sequence of appearances (the average credible announcement to date). This latter sequence may or may not have a limit.[11] Because of the infinite horizon and the nature of the updating process, if the limit does exist for a given player, then the forecasts made by the other players about this player will also converge, and to the same point. In this case, we specify that entering the concrete action game, the beliefs held by the other players about this player are also this same point in the strategy space. In this way, the conversation is a model of belief formation. If the appearance does not converge, then the appropriate forecast will not converge either,

---

[8]   We assume that players only care about payoffs up to some arbitrarily small constant $\varepsilon$, either because they cannot perceive finer differences or because they are indifferent over this range.

[9]   We make the standard assumptions on the action game so that a best response always exists.

[10]   In particular, continually starting over inhibits convergence, in which case, the player has no influence on the ultimate course of the discussion. This is never optimal, as shown below.

[11]   If no credible announcements were made after some finite stage, this is taken to mean that the limit does not exist. However, as above, we may assume that this does not occur.

and beliefs remain open for the time being. Of course, it may be true in general that appearances have a limit only for some (possibly empty) subset of the group of players.

If the appearances of all players converge, then we say that the conversation itself converges. However, in this case, every player continues to make credible announcements, and hence, at the limit, these announcements must be near the best responses to the actions stated by the other players, and hence to the limits of the other players. Since by definition, the latter are the beliefs held by the given player upon entering the action game, his or her limit must be an action that is (near) the best response to his or her beliefs and is therefore one optimal strategy to pursue in the action game. Therefore, we may assume that this limit action is indeed chosen, validating the beliefs of the other players. Of course, since this is true for all players, the limits must be mutual best responses, and thus, the play arrives at a Nash equilibrium. This is Theorem 1 below.

We next turn our attention to the question of optimality in the cheap talk stage of the overall game. Stepping back for a moment, we consider the question of whether or not to participate in the conversation at all, given the opportunity to do so. Since there is a natural language with which to communicate, any player can initiate a conversation. Whether or not they choose to participate, other players will hear and be influenced by the announcements of this player. Therefore, if they do not also make announcements, this player (or players) will have free reign to drive beliefs toward the equilibrium of their choosing (by announcing it ad infinitum). Since this outcome is at least weakly bad for other players, it cannot hurt them to also join in the conversation and attempt to guide the discussion in a direction favorable for them. For instance, in the Battle of the Sexes game, played between one man and one woman, Player 1 conversing with himself will continuously announce the equilibrium that he prefers. Entering the action game, the other player believes these announcements and best responds to them, so that the play will in fact be at that equilibrium. In this case, it would have been a good idea for Player 2 to at least try to promote her favored outcome, that is to participate in the conversation. Thus, we may assume, without any loss of generality, that all players converse.

Players do not know the cheap talk strategies employed by their opponents (if they did, we should instead be modeling what occurred before this conversation in order for that knowledge to be gained), so these players must consider all strategies to be possible. Thus, if a cheap talk strategy for one player never performs better (in terms of the payoffs ultimately realized in the action game, of course) than another competing strategy and does strictly worse against at least one possible strategy profile of the opponents, then the original strategy should be discarded as suboptimal. Anything that is not weakly dominated is optimal.[12] This is intentionally a broad definition of a strategy; it is meant to be as loose as possible and yet at least minimally capture the requirements of optimality. Theorem 2 below proves that if all players employ communication strategies that are optimal in this loose sense, then the conversation must converge to a stably efficient equilibrium of the game. This class of equilibria, defined below, is essentially those Nash equilibria for which no coalition can break away and, on their own, force the other players to follow them to some other equilibrium that is preferred by the coalition. In two person games with distinct payoffs (a property that holds generically), this result is equivalent to Pareto optimality.

## 4. Model

Consider a game **G** with $n$ players and finite action spaces $S_i$ for $i = 1, ..., n$.[13] Payoffs are given by $u_i$ for $i = 1, ..., n$. It will be simplest to think of **G** in normal form. **G** is played exactly once, though **G** itself may be a repeated game. Before this happens, there is a **conversation** C(**G**), defined as follows. Each player begins the pregame conversation with a totally mixed prior **forecast** $\pi_i = \pi_i^1 \in \Delta(S_i)$ about his or her behavior. The forecasts are common knowledge among all the players. At each

---

[12] Naturally, since full rationality is assumed, we could endlessly iterate the process, but there is no need.
[13] The assumption of finiteness can be weakened.

round $t = 1, 2, 3...$ of the conversation, player $i$ announces $m_i^t \in S_i$. The announcements are made simultaneously by all players in each round.[14]

Let $NE(G) \subseteq \underset{i=1}{\overset{n}{\times}} \Delta(S_i)$ be the set of Nash equilibria of **G**, and define $E_i \subseteq S_i$ by:

$$E_i = \{s_i \in S_i \mid \exists \sigma \in NE(G) \text{ with } s_i \in supp(\sigma_i)\}.$$

This set constitutes the self-committing actions for player $i$. At $t = 1$, any $m_i^1 \in E_i$ is said to be **credible**. If $m_i^1$ was credible, then we define:

$$\pi_i^2 = (T\pi_i^1 + m_i^1)/(T+1)$$

for some fixed $T$, which can be chosen to be large relative to the scale of the strategy space and payoffs in the underlying game. This captures the slow updating process of prior forecasts by credible announcements. In a similar fashion, the **appearance** is given by $p_i^2 = m_i^1$. If the initial announcement was not credible, then the forecast is not updated, and the appearance is undefined. Recursively, we now define $m_i^t$ to be credible when:

$$m_i^t \in \varepsilon BR_i(\underset{j \neq i}{\times} q_j^t) \text{ with } q_j^t = \pi_j^t \text{ or } p_j^t \forall j,$$

where $\varepsilon BR_i(\sigma_{-1})$ denotes:

$$\left\{ s_i \in S_i \mid \max_{s_i' \in S_i} u_i(s_i', \sigma_{-1}) - u_i(s_i, \sigma_{-i}) < \varepsilon \right\}$$

for some arbitrarily small $\varepsilon > 0$. If $m_i^t$ is not credible,[15] then $\pi_i^{t+1} = \pi_i^t$ and $p_i^{t+1} = p_i^t$. If $m_i^t$ is credible, then we define:

$$\pi_i^{t+1} = ((T + t - 1)\pi_i^1 + m_i^1)/(T + t) \text{ and } p_i^{t+1} = ((t-1)p_i^t + m_i^t)/t.$$

Say that player $i$'s appearance *converges* if player $i$ never entirely stops making credible announcements and if $\lim_{t \to \infty} p_i^t$ exists. If this happens, it is clear that $\lim_{t \to \infty} \pi_i^t$ also exists and is the same; call it $b_i$ for the belief about player $i$. If the limit exists for all players, then the conversation converges. In this case, we assume that beliefs after the conversation and entering **G** are given by $\mu_i = \underset{j \neq i}{\times} b_j$.

**Definition 1.** *An **acceptable equilibrium** (of **G**) is a profile $\sigma \in \underset{i=1}{\overset{n}{\times}} \Delta(S_i)$ such that $\sigma = b$ for some belief vector $b$ resulting from a convergent conversation starting at some prior forecasts $\pi$; the set of acceptable equilibria is denoted $AccE(G)$.*

**Theorem 1.**

1. $NE(G) \subseteq AccE(G)$
2. $AccE(G) \subseteq \varepsilon NE(G)$

**Proof.** (1) Let $\sigma \in NE(G)$, and consider prior forecasts $\pi$ very close to $\sigma$. By the definition of a Nash equilibrium, any $s_i \in supp(\sigma_i)$ is in $\varepsilon BR_i(\pi_{-i})$. Now, let the players announce actions in the support

---

[14]  Sequential announcements lead to a forced asymmetry regarding who speaks when. The effects of this generalized first-mover advantage are irrelevant for the present discussion.

[15]  Unless player $i$ has only one possible credible announcement, as discussed in Section 3.

of $\sigma$ in such a way as to match as nearly as possible the actual distribution prescribed by $\sigma$. Initially, all these actions will be credible as stated. Of course, the forecasts will change over time, but since the updating process is slow and the cheap talk announcements are matching the given distribution, the forecasts will always stay near $\sigma$. Hence, the actions in the support of the announcements will remain credible forever. In this manner, $\lim_{t\to\infty} p_i^t$ exists $\forall i$, and moreover, $\lim_{t\to\infty} p_i^t = \sigma_i$. Thus, $\sigma$ is indeed an acceptable equilibrium.

(2) If $\sigma \in AccE(G)$ and so is the limit of a convergent conversation, it must be that all $s_i \in \text{supp}(\sigma_i)$ are credibly announced infinitely often during the preplay cheap talk stage.[16] Since in the limit, both the forecasts and the appearances are arbitrarily near $\sigma$, each such $s_i$ must be in $\varepsilon BR_i(\sigma_{-1})$, and therefore, $\sigma_i \in \varepsilon BR_i(\sigma_{-i}) \forall i$. $\square$

Among other things, this result justifies the possibility that after a convergent conversation, players both rationally and self-consistently hold the beliefs that are given by the model. Theorem 1 in some sense clarifies the relationship between cheap talk (as has been modeled here) and Nash equilibrium. If the communication is meaningful, that is if the cheap talk has a limit, then it must lead to a Nash outcome. Of course, there is no guarantee that the conversation will converge, and it is quite possible that it will not.[17] Furthermore, no Nash equilibrium, even if inefficient, can yet be ruled out. Something stronger than an acceptable equilibrium is required.

We next turn to defining the appropriate efficiency concept in this setting.

**Definition 2.** *Call $\sigma \in NE(G)$ **directly attainable** from $\sigma' \in NE(G)$ by the coalition S if $\sigma_s$ is a Nash equilibrium in the induced game fixing all players outside of S to play as in $\sigma'$, and if also $\forall i \notin S$, we have $u_i(\sigma_i, \sigma_S, \sigma'_{-i,S}) > u_i(\sigma'_i, \sigma_S, \sigma'_{-i,S})$.*

This is a strenuous definition: the first condition asks that the members of $S$ be able to "jump" to $\sigma$ from $\sigma'$, and the second condition requires that once they have done so, they can force the rest of the players to follow them.

**Definition 3.** *Call $\sigma \in NE(G)$ **attainable** from $\sigma' \in NE(G)$ by the coalition S if there is a chain of equilibria, each directly attainable by S, leading from $\sigma'$ to $\sigma$; if also, $\forall i \in S$ $u_i(\sigma) > u_i(\sigma')$; and if finally, there is no similar such chain (for any coalition) leading away from $\sigma$.*

These are once again fairly strict requirements. The second one states that all members of $S$ must strictly prefer the new equilibrium, and the third states that the new equilibrium itself is immune to these sorts of deviations.

**Definition 4.** *A Nash equilibrium of **G** is **stably efficient** if nothing is attainable from it; the set of these equilibria is denoted $StEff(G)$.*

By considering the grand coalition of all players, it is clear that an equilibrium exhibiting stable efficiency will tend to be efficient. In games with distinct payoffs, no singleton coalitions can ever attain alternate equilibria (this follows from the first condition of the first definition), and hence, in two-person games, stable efficiency is generically equivalent to efficiency. It is clear that stably efficient equilibria always exist (since whatever is attained must itself be stably efficient). In most games, efficiency and stable efficiency will coincide, but when they do not, it is important that we use the latter concept. Stable efficiency is related to the coalition-proof concept introduced by [21], but is more

---

[16]  In particular, since the conversation converges, there must be some round after which nobody ever cleans their slate and starts over.

[17]  Consider, as one example, fictitious play in the rock-paper-scissors game.

farsighted in that it looks at the full implications of a coalitional deviation; it turns out that neither definition is a refinement of the other.

Recall that a cheap talk strategy is **optimal** if it is not weakly dominated.

**Definition 5.** *An **agreeable equilibrium** (of G) is a profile $\sigma \in \overset{n}{\underset{i=1}{\times}} \Delta(S_i)$ such that $\sigma = b$ for some belief vector b resulting from a convergent optimal conversation starting at some prior forecasts $\pi$; the set of agreeable equilibria is denoted $AgrE(G)$.*

**Theorem 2.**

1.  $StEff(G) \subseteq AgrE(G)$
2.  $AgrE(G) \subseteq \varepsilon StEff(G)$

**Proof.** (1) Consider $\sigma \in StEff(G)$, and let the prior forecasts $\pi$ be very close to $\sigma$. Since the forecasts favor $\sigma$ so heavily, the only way that another equilibrium can ever be reached during the conversation is if it is directly attainable or the result of a chain of directly attainable equilibria. Thus, all of the players know that these are the only feasible outcomes, and in fact (see the strategies below), they can be reached in a conversation. However, since $\sigma$ is stably efficient, it is not possible for any player (as a member of any coalition) to be sure that by deviating from one of these alternates, a superior payoff can be achieved. It must be the case that either not all members of the coalition will profit by the switch (in which case, those who do not profit will not participate in the deviation) or if they do, that then, there is another coalition that can profitably and successfully deviate away from this new point. Of course, it is possible that one's payoff will be increased by attempting to switch equilibria, but there will always be circumstances in which it is not profitable. Thus, there is no strategy that weakly dominates the strategy "emulate $\sigma$", which is always available due to the prior forecasts. This implies that one optimal strategy for all players is to follow $\sigma$, and the result of this will be that the conversation converges with $\sigma$. There may be other optimal strategies, and there may be other possible results to the conversation; however, this is sufficient to show that $\sigma \in AgrE(G)$, as desired.

(2) Suppose that a conversation is converging toward an equilibrium $\sigma$ that is not stably efficient (even up to $\varepsilon$-indifference). If there is just one coalition that can attain a superior equilibrium for itself, it can pursue the following strategy: (a) Erase its current appearance and start over, and then, (b) announce the actions that lead to the first equilibrium along the chain. If all members of the coalition have done likewise, then they will be able to credibly repeat those announcements in the next round, since these are mutual best responses given the forecasts near $\sigma$ for the other players. If the other members have not done this, each individual can start over again and try once more. If eventually they coordinate, then they can continue to make these announcements indefinitely. At some point, the forecasts and appearances will then be very close to this new equilibrium, and the only credible choice for the other players will be to switch to it as well (this follows from the definition of directly attainable). They can continue in this fashion until the final equilibrium in the chain, where the process will conclude (by the argument in Part (1) above).

Of course, this attempted deviation will not always work, but it is safe in that either it works (that is, all members of the coalition coordinate) and a higher payoff is realized or it does not and the conversation stays at $\sigma$ instead. Therefore, the deviation strategy weakly dominates the "emulate $\sigma$ and stay where you are" strategy. Since this is true for all members of the coalition, optimality implies that all of them will attempt to force the switch to the preferred attainable equilibrium, and with probability one they will eventually coordinate (since they always have the opportunity to start over). Therefore, $\sigma$ was not in fact an agreeable equilibrium.

Similarly, if there were several coalitions that could attain superior equilibria, each member of each coalition can start over at each round and attempt to coordinate with his or her coalition. Any player who is a member of several coalitions or who has a choice between attainable equilibria can randomize between these possibilities. If the player puts almost all weight on his or her individually preferred

outcome among all these choices and spreads $\sigma(\varepsilon)$ weight across the others, then this will be $\varepsilon$-optimal, but will at the same time guarantee that with probability one, coordination takes place at some point. This weakly dominates "emulate $\sigma$" because either the conversation converges to $\sigma$ anyway (though this never actually happens with optimality), or another coalition coordinates (which could not be helped), or one of the attempted coalitions coordinates first (which increases payoffs). Therefore, once again, no optimal conversation will remain at $\sigma$, and thus, it could not have been agreeable.    □

The intuition behind Part (1) is particularly simple in two player games. In this case, given a strong prior forecast, either player can insist on the original equilibrium $\sigma$ for longer than the other player can credibly hold out against it (by the definition of Nash). Therefore, both players must optimally be able to get at least their payoff from $\sigma$. However, since $\sigma$ is efficient, this means that both players get exactly this payoff under any optimal strategies, and thus, staying at $\sigma$ itself is as good as anything else. The examples in the next section serve to illustrate the mechanisms behind both the definitions and the proof of the theorem. It should be pointed out that in most specific cases, very little of the somewhat complex machinery developed above is necessary or applicable; the process is often hopefully quite natural and intuitive.

## 5. Examples

The most obvious example of an equilibrium selection problem is posed by the following coordination game:

|       | *A*   | *B*   |
| ----- | ----- | ----- |
| *A*   | 2,2   | 0,0   |
| *B*   | 0,0   | 1,1   |

Of the three Nash equilibria in the game, only one is efficient. There is also an inefficient equilibrium, and this type of coordination problem comes up often in many contexts—including viral pandemics (see, e.g., Jnawali et al. 2017 [22]). Although in scenarios without communication, it is possible for (B,B) to occur, Theorem 2 implies that the efficient equilibrium $(A, A)$ is the only possible outcome after rational nonbinding communication takes place among the players, no matter the prior forecasts. This is easy to see if either of the forecasts puts significant weight on $A$. In that case, the other player can credibly repeatedly announce $A$ as the best response and, in this manner, eventually force the only credible announcement by either player to be $A$. Since this yields the highest possible payoff, it is optimal, and the conversation will converge to $A$.

If instead the prior forecasts are both heavily skewed toward $B$, then each player can reason as follows: "If I announce $B$, we will be stuck there forever, and I will get a payoff of one. If I announce $A$, there is some chance that my opponent will announce $B$, in which case, we will get stuck, and I will receive one. However, there is also some chance that my opponent will announce $A$. If we both continue to do this, these will remain credible announcements (since they each best respond to the other's appearance), and we will converge to the efficient equilibrium, delivering me a payoff of two instead of zero or one. I can always go back to announcing $B$ and force that equilibrium (or start over altogether), so there is no risk of ending up at the really inefficient mixed equilibrium. Since there are no instantaneous payoffs lost from miscoordination along the way, the only possible optimal strategy is for me to announce $A$."

Both players are rational, so they will in fact both announce $A$ at all rounds of the cheap talk communication, and the conversation will end up converging to the efficient equilibrium. Given that the forecasts were heavily skewed toward $B$, it may be a long time before the two players have truly convinced each other of their intention to play $A$, but they have all the time in the world and every reason to make use of it. If we looked instead at the pure coordination game in which $(A, A)$ also yields payoffs of one to each player, the analysis is slightly changed. If the prior forecasts lean toward either of the symmetric and efficient pure equilibria, the conversation will converge in that direction. However, if the priors miscoordinate just right (for example, they are completely uniform

for both players), it will be necessary for both players to randomize their initial announcement. If they coordinate at that point, successful convergence follows. If not, they simply clean their slate, start over, and try again. At some point, they must (that is, with probability one) both choose the same action (this is why it is necessary to randomize rather than to try to coordinate in some deterministic pattern), and then, they are done.

A less clear-cut example with a unique efficient equilibrium is found in the following version of the "stag-hunt" game:

|   | S | R |
|---|---|---|
| S | 5,5 | 0,4 |
| R | 4,0 | 3,3 |

Here, the unique efficient equilibrium involves choosing a risk-dominated action, perhaps making it more difficult to reach. Allowing communication, however, will afford the players an opportunity to convince each other that it is safe to play action $S$. [23] has argued to the contrary that cheap talk may not help in this game. His reasoning is that since each player would prefer the other to take action $S$, they should each attempt to convince the other player to choose it. The way to do this is by claiming that you yourself are also going to pick $S$. Therefore, hearing the other player announce $S$ should be discounted as purely manipulative and ignored.

It seems that Aumann's argument is not self-evident, at least when there is an unlimited chance to communicate. Rational players know that they will eventually agree on a Nash equilibrium; there is zero probability of suckering the other player or miscoordinating. At this point, it comes down to a choice among equilibria. Knowing this perfectly in advance, if a player announces $S$, it must be because he or she is hoping to eventually end up at the efficient equilibrium, that is to end up playing $S$. It is, after all, the best response at that point. In any case, the data clearly support the idea that allowing preplay messages increases the probability of observing the efficient, but risk-dominated equilibrium; see Charness (2000) and Miller and Moser (2004) [24,25].

We turn our attention next to the Battle of the Sexes, which is not at all a game with common interests:

|   | F | B |
|---|---|---|
| F | 2,1 | 0,0 |
| B | 0,0 | 1,2 |

In this case, it is not immediately obvious that even with communication, efficiency can necessarily be achieved. If the prior forecasts favor either one of the pure equilibria, then the player who prefers that equilibrium will be able to credibly "insist" on it, and it will be the ultimate limit of the conversation. If the forecasts are balanced, however, neither player can be assured of getting his/her preferred outcome. Insisting on it whenever possible may lead the conversation to converge toward the inefficient mixed equilibrium, which is worse for both players. Therefore, this strategy is not optimal. If instead, the players "yield" to the other player with some extremely small probability at each round, this will always be achieved within $\varepsilon$ of any other strategy, and since it always leads to one of the efficient equilibria, it weakly dominates the strategy by a player that forever insists on getting his or her way. Thus, under this scenario, the players are behaving optimally and can achieve efficiency with certainty.

As a final example, we turn to games with three players in order to explain some of the added complexity that arises. First, consider the following game in which the matrix player's payoffs are listed last:

|   | L | R |   |   | L | R |
|---|---|---|---|---|---|---|
| U | 0,0,10 | −5,−5,0 |   | U | −2,−2,0 | −5,−5,0 |
| D | −5,−5,0 | 1,1,−5 |   | D | −5,−5,0 | −1,−1,0 |
|   | *A* |   |   |   | *B* |   |

This game has two pure Nash equilibria, namely $(U, L, A)$ and $(D, R, B)$, only the first of which is efficient. The second equilibrium is directly attainable from the first through a coalition of the row and column players, but it is not fully attainable because they enjoy a lower payoff in this equilibrium. Thus, the first equilibrium is stably efficient (and hence, the second, dominated one cannot be) and will be the result of rational communication. Nevertheless, since the row and column payoffs would be higher at the intermediate point along the chain fixing the matrix player at $A$, the original efficient equilibrium is not coalition-proof. Now, modify the payoffs slightly:

|   | $L$ | $R$ |   |   | $L$ | $R$ |
|---|---|---|---|---|---|---|
| $U$ | $2, 2, 10$ | $-5, -5, 0$ |   | $U$ | $-2, -2, 0$ | $-5, -5, 0$ |
| $D$ | $-5, -5, 0$ | $1, 1, -5$ |   | $D$ | $-5, -5, 0$ | $3, 3, 0$ |
|   |   | $A$ |   |   |   | $B$ |

Only the equilibrium payoffs have been changed, but the analysis has been affected greatly. Both pure equilibria are now efficient, but for exactly the reasons outlined above, only the second one, $(D, R, B)$, is stably efficient and can be the result of cheap talk. On the other hand, the original equilibrium is now coalition-proof, showing the discrepancy between the two concepts.

One of the (unavoidable) limitations of this model is that it can say nothing about zero-sum games, except that communication can only converge to a Nash equilibrium. Other games in which all equilibria are efficient, and so for which Theorem 2 is vacuous, are games with a unique Nash equilibrium. These include matching pennies, rock-paper-scissors (where many of the convergence problems of fictitious play show up), and the game-theoretic standby of the prisoner's dilemma. Of course, we cannot expect that simple communication would lead to cooperation, a strictly dominated strategy. We have assumed throughout that there is only a single (though unlimited) chance for the players to talk for playing a game. If **G** is a repeated game and the players have a full conversation between each stage, then optimal speech should lead to efficient outcomes all along the extensive form game tree, both on and off the equilibrium path. This gives rise to the difficult problem of finding renegotiation-proof equilibria[18].

## 6. Conclusions

Coordination games of various forms, from actual rendezvous games to super-modular games and complementarity games, have received increasing attention in the game theory literature. Most equilibrium selection in such games, however, has been relatively informal, appealing to such concepts as focal points, initial conditions, or competition (essentially an evolutionary argument). Cheap talk, meaning costless and nonbinding preplay communication, has presented an intuitively pleasing method for formally attacking the equilibrium selection problem. The model of *conversations* presented here attempts to provide one possible resolution to this question of equilibrium selection, as well as to the even older question of justifying the Nash equilibrium concept.

The model assumes that players meet for the first time and communicate in order to allay their uncertainty about the future actions of their opponents. Since they have no knowledge of the cheap talk strategies used by the other players, we do not look for an actual equilibrium of the extended game. Instead, we look for all outcomes that could reasonably occur as the result of rational communication on the part of the players. Messages are defined to be credible in the context of a particular conversation. If at the end of a conversation, a player has put forward a consistent and credible appearance, this is assumed to in fact be the other players' belief about his or her future actions. From this base, it is proven that meaningful communication (that is, in which there is convergence) must end up at a Nash equilibrium. This is a partial justification for the Nash concept. It is then proven that optimal

---

[18]   See, for example, the survey paper by Bergin and MacLeod (1993) [26].

communication, meaning that all players make strategic and rational announcements, leads to the deselection of inefficient equilibria.

A strength of the paper is that it gives a decisive answer to these two issues within the context of a single model. It also applies to games with more than two players or that do not necessarily exhibit common interests. There are, however, several qualifications to the model. First, the results do not prove that convergence must take place, only that if it does, then it takes a certain form. Secondly, since by no means all applications allow the possibility for preplay communication, this cannot be a general justification for the Nash concept. Indeed, this also potentially predicts a distinction between environments where one would expect Nash equilibrium to obtain versus others where one would not necessarily expect it. Finally, the model does put restrictions on the belief formation process, in that it requires some very small amount of faith to be put in credible announcements, at least over the long run. Note that this is not a departure from full rationality; traditional models have simply left this process unmodeled. There are also a number of possible relevant extensions of this model, notably to correlated equilibrium and to introducing a stochastic element in the conversation.

Calvin Coolidge once wisely said, "It is better to remain silent and be thought a fool than to speak and prove it." However, that applies only to fools: the moral of this paper is, "It is worse to remain silent and only be supposed rational than to speak and confirm it."

**Funding:** This research received no external funding.

**Acknowledgments:** I would like to thank Max Amarante, In-Koo Cho, Lynn Conell-Price, Peter Diamond, Glenn Ellison, Ehud Kalai, Elizabeth Murry, Tom Palfrey, Lones Smith, Radhanjali Shukla, and the seminar participants at Caltech, MIT, Penn State, Rochester, and the 17th Arne Ryde Symposium for valuable comments. All remaining shortcomings are mine only.

**Conflicts of Interest:** The author reports no conflict of interest.

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
