# Peer review of "Valuable Cheap Talk and Equilibrium Selection"

_games, doi:10.3390/g11030034_

Round 1

Reviewer 1 Report

My comments must be addressed.

  1. Authors are highly encouraged to include line number in the manuscript.
  2. In Introduction section, please define the Nash Equilibrium and cite the appropriate book/ paper. Such as

Osborne, M. J. (2004). An introduction to game theory (Vol. 3, No. 3). New York: Oxford university press.

  1. In literature review section, author must clarify that why this research study is imminent.
  2. In examples section, author mentioned that (A, A) is efficient equilibrium. However, I can see two equilibria. Is it possible both player end up adopting the socially suboptimal strategy? Explain this case. Please look at the research paper below to elaborate this part and cite the paper appropriately.

https://currents.plos.org/outbreaks/article/strategic-interactions-in-antiviral-drug-use-during-an-influenza-pandemic/

  1. The conclusion section is very long. Authors are highly encouraged to reduce the conclusion section.

Author Response

Thank you for reading the paper and for your comments.

  1. I have added line numbers as requested.
  2. Good point! - done
  3. I'm not certain exactly what you mean by "imminent" but I think I understand your point (that I should go beyond simply describing the previous literature and state the value-added of this paper). I have tried to do this now, toward the end of Section 2 on p.4 (and see point 5 below).
  4. Sorry if this was unclear - yes there is a second pure strategy NE which is inefficient, and that could indeed occur, but not if players optimally communicate beforehand (exactly as per Theorem 2 in the paper). I have tried to clarify this point, and have added the recommended citation.
  5. The conclusion is only just over one page, and it seemed intellectually necessary to transparently discuss the limitations and drawbacks of the paper. There is also some of the discussion here (re the marginal contribution of this model in particular) that you requested in your point 2. However I have deleted a sentence on experimental verification of the results, and if there are other specific areas that you think I should cut then I would be happy to consider doing so.

Reviewer 2 Report

This is the report of the paper “Valuable Cheap Talk and Equilibrium Selection” (games-845514)

Summary and general comments

The paper starts with the observation that there is a tension between the intuitive sense that players who are allowed to engage in costless communication before playing a game would be foolish to agree on an inefficient equilibrium and that, at the same time, however, such pre-play communication has been suggested as a rationale for expecting Nash equilibria in general. I have a major comment on this motivation which I explain later.

Overall, this paper presents a model of cheap talk that tries to resolve this tension. The model is described as “plausible” by the authors and I think that the infinite nature of communication makes it less plausible than is argued by the authors.

The setting is quite special in that players are assumed to have an unlimited opportunity to send messages before playing an arbitrary game. This period of cheap talk communication is assumed to be infinite and is named a conversation. Using an extension of fictitious play beliefs, the authors make assumptions concerning which messages about future actions are credible and hence contribute to final beliefs. The authors describe the assumptions as minimal and I am not sure that these are indeed minimal. The authors argue that meaningful communication among players leads to a Nash equilibrium of the action game. Within the set of NE, efficiency then turns out to be a consequence of imposing optimality on the cheap talk portion of the extended game. The authors further argue that the finding contrasts with previous 'babbling' results but they do not explain why. Overall, it is very hard to follow the intuition as there are no graphs.

I find that the main motivation of the paper, in my eyes, is to explain the role of conversations before equilibrium selection which is really important. I think that it is crucial to understand the impact of freeplay communication on equilibrium selection so I would be interested in seeing a revised version of this paper. However, the very intangible aspects of communication in its formal description makes me skeptical that a revised version of the paper could grasp what the authors want it to grasp. I am worried that the model is far from a model that would describe how conversations occur. I try to find ways to distinguish this conversation from how smart algorithms would function and I can’t distinguish the two. I think the authors should try to rephrase the introduction to avoid these kind of remarks. I regret that some important aspects of the paper are not discussed in a clean fashion. I hope my comments below can be helpful.

Comment 1

The first sentence in the abstract and the main point that motivates the paper is not intuitive at all, and it goes against experimental evidence on public good games in which pre-play communication leads to not choosing the NE. It would advise to rephrase the abstract so that the abstract provides more accurate information about what the paper does. Put simply, a Nash equilibrium is not a contract.

Similarly, I am not sure that the contract example in the introduction is appropriate and truly captures the content of the paper. If the authors want to keep it, I would be interested in having it more connected to the paper because the connection seems too obscure at this stage. It’s to be a reasonable to completely rephrase this or remove it.

Comment 2

The motivation is not tied in a clear fashion to coordination problems when in fact it seems that the paper is restricted to this question. I would suggest making the paper clearer in this respect or explain what the scope of the paper is if it covers a broader range of games but my reading of the paper is that it is not the case. It seems that the example in Section 6 should be much more central then where it stands now. This restricts the scope of the paper but it makes it clearer.

Comment 3

I have questions about the assumption behind the fact that the forecast is common knowledge to the players. It would be useful to define why that assumption does not make the game tautological. Page 5 provides an explanation in the footnote which says that it is not necessary for the results that the prior are mixed. However, at the beginning of the section, it says that expectations can be some vague initial ideas emerging from focal points or conventions.  

Comment 4

The paper is not tied at all to the experimental evidence on the effect of preplay communication in coordination games and it gives the impression that the authors are loosing an opportunity to motivate the results. The results of the paper, although formal here, are not new to experimentalists. This cannot be ignored. The authors should cite the relevant experimental economics literature, which would give more depth to the paper.

It could also include a references to the results on communication with leaders as one player must start the conversation: See for example:

Although not directly related, the authors should also look at the literature on leadership as it is about sequential communications and it could help the authors find ways to make their theoretical claims more tangible. See For coordination games: Jordi Brandts, David J. Cooper It's What You Say, Not What You Pay: An Experimental Study of Manager-Employee Relationships in Overcoming Coordination Failure Journal of the European Economic Association, Volume 5, Issue 6, 1 December 2007, Pages 1223–1268 and for PPG: Béatrice Boulu-Reshef, Charles A. Holt, Matthew Rodgers and Melissa Thomas-Hunt (2020), "The impact of leader communication on free-riding: An incentivized experiment with empowering and directive styles", Leadership Quarterly, 31(3): 101351 for public good games (although that may be far).

Minor comments

Page 4: “to the next section”. There is a letter missing.

Page 16: With all my due respect to Aumann, I would soften slightly the wording. But the authors are free to leave it as is. It is just an suggestion.

Page 16: I would not say “perfectly safe”.

Page 16: what “data” do the authors refer to in the last paragraph?

There are sentences that are not clear as for example: “If they coordinate at that point, fine. If not, they simply clean their slate, start over, and try again.” Or “Another approach that will destroy the babbling equilibria is to assume an arbitrarily small but positive cost to sending messages”.

Author Response

First of all, thank you for taking the time to carefully read the paper and for your many helpful comments. I'm glad that you find the topic to be really important, as do I.

You are of course correct that infinite conversations are not plausible in the real world, but I see that as a modeling device - in general it will converge rapidly (although this will depend on the exact belief updating weight chosen, so there is a trade-off involved and I wanted to allow as much flexibility as possible here). I have tried to justify this a bit further on p.2 ("Realistically...").

As to whether the assumptions imposed here are minimal, that is naturally in the eye of the beholder. I will point out that my main theorems go in both directions to some extent, so I doubt that weaker assumptions would achieve the same results. In some sense I believe that these assumptions are roughly what is required to formalize existing intuitions about the link between communication and Nash equilibria (rather than, say, trying to model the implications of the most realistic forms of actual communication, which is a valuable but distinct endeavor).

Finally, the distinction with babbling results was explained at the bottom of p.2.

Responses to comments:

  1. The first sentence of the abstract was meant to refer to choosing an inefficient choice within the set of equilibria (which seems extremely intuitive to me, and is backed by experimental evidence). However I realize it was slightly ambiguous as written, so I have tried to make this clearer - thank you. Meanwhile I agree of course that a key characteristic of NE is that they are not contracts, although I found the analogy helpful. However I defer to the reader on this and have removed the opening sentences so as not to distract from the main point. Finally that's a good point that in some experiments cheap talk leads to more efficient but non-Nash outcomes - I have added a discussion of this possibility at the end of Section 2 (p.4) but I don't believe it negates my theoretical results.
  2. Sorry that this was unclear -- coordination games are of course a natural motivating example for the discussion of inefficient equilibria, but the scope of the paper (especially the first theorem) is indeed much broader. I have added a sentence to this effect on p.2
  3. Good question. The difference (as I see it) is that forecasts apply to everyone and change (if at all) at a societal level, whereas beliefs in an individual game apply only to those particular players. I have added some discussion of this at the end of the first paragraph of section 3 (p.5). In any case I don't see how this would make the theorems tautological? 
  4. I'm not sure if there is a missing citation here or if you simply restarted your paragraph on leadership. On the one hand of course I agree that in some ways this is simply formalizing what is already known to experimentalists and is indeed highly intuitive (hence the first sentence of the abstract). Formalization is useful, if only to understand the assumptions and boundaries of the results. I had already cited some of the experimental literature, and I have added the two that you suggest (which are both on target - thank you). All that being said, as I mentioned above the goal is primarily to elucidate the casual theoretical justifications often given for NE, rather than to realistically model actual communication paradigms, hence lab experiments are not directly relevant.

Responses to minor comments: thank you, I have edited the relevant points in the text including softening the wording and adding two citations to experimental evidence on stag-hunt games (contra Aumann), which may also help with your point 4 above.

Reviewer 3 Report

The paper is well written.

The subject to be studied is well described and analysed.

As far as I know, the results are original.

Some text errors must be corrected. For example:

page 4, line - 8: replace "the" by "the"

page 6, line -9: replace "need" by"needs"

Author Response

Thank you! - I'm glad you liked the paper. Good catch on the typos, and I have made the changes.

Round 2

Reviewer 2 Report

Dear authors,

Thank you for the responses that were very clear. The presentation and motivation of the paper are now clearer. The inclusions you mention in your response letter do make the paper clearer. 

Kind regards 

Author Response

Thank you - I'm glad it helped